# The Myeloid Biomarker *MS4A6A* Drives an Immunosuppressive Microenvironment in Glioblastoma via Activation of the PGE2 Signaling Axis

**DOI:** 10.3390/ijms27010058

**Published:** 2025-12-20

**Authors:** Jianan Chen, Qiong Wu, Anders E. Berglund, Robert J. Macaulay, James J. Mulé, Arnold B. Etame

**Affiliations:** 1Department of Neuro-Oncology, H. Lee Moffitt Cancer Center and Research Institute, Tampa, FL 33612, USA; jianan.chen@moffitt.org (J.C.); qiong.wu@moffitt.org (Q.W.); 2Department of Quantitative Health Sciences, Division of Computational Biology, Mayo Clinic, 4500 San Pablo Road South, Jacksonville, FL 32224, USA; berglund.anders@mayo.edu; 3Departments of Anatomic Pathology, H. Lee Moffitt Cancer Center and Research Institute, 12902 Magnolia Drive, Tampa, FL 33612, USA; robert.macaulay@moffitt.org; 4Department of Immunology, H. Lee Moffitt Cancer Center and Research Institute, 12902 Magnolia Drive, Tampa, FL 33612, USA; james.mule@moffitt.org

**Keywords:** Glioblastoma, tumor-associated macrophages, *MS4A6A*, single-cell transcriptomics, immunosuppressive tumor microenvironment

## Abstract

Glioblastoma (GBM) remains one of the most lethal brain tumors, characterized by extensive immune evasion and a macrophage-dominated tumor microenvironment (TME). However, the molecular determinants governing tumor-associated macrophage (TAM) states and their immunoregulatory functions remain poorly understood. We integrated bulk- and single-cell transcriptomic datasets (TCGA, CGGA, Ivy GAP, and Brain Immune Atlas) to systematically characterize the expression, prognostic relevance, and immune contexture of the myeloid biomarker membrane-spanning 4-domain A6A, *MS4A6A*, in GBM. Differential expression, survival, and pathway enrichment analyses were performed. Single-cell mapping and CellChat modeling delineated *MS4A6A*-associated TAM subpopulations, intercellular communication networks, and ligand–receptor signaling dynamics. Spatial transcriptomic validation and pharmacogenomic modeling were conducted to assess anatomic enrichment and therapeutic vulnerabilities. High *MS4A6A* expression predicted unfavorable survival and correlated with increased stromal and immune infiltration. Single-cell analyses localized *MS4A6A* predominantly to TAMs, especially Regulatory- and Ribo-TAM states enriched for antigen presentation, T-cell regulation, and ribosomal biogenesis pathways. CellChat analysis revealed that *MS4A6A*-high TAMs exhibited markedly enhanced communication with CD4^+^ T cells and Tregs through upregulated PGE_2_–PTGER2/PTGER4, PECAM1–CD38, and THBS1–CD36 signaling axes, implicating *MS4A6A* in prostaglandin-driven immune suppression. Spatial profiling confirmed preferential localization of *MS4A6A* within perivascular and angiogenic niches. Pharmacogenomic prediction indicated that *MS4A6A*-high tumors were more sensitive to ERK, mTOR, and CDK4/6 inhibition. *MS4A6A* defines a macrophage-centered, immunosuppressive ecosystem in GBM, mediated by the activation of the PGE_2_ signaling axis. These findings position *MS4A6A* both as a prognostic biomarker and as a potential therapeutic node linking myeloid reprogramming to actionable pathway vulnerabilities in glioblastoma.

## 1. Introduction

Glioblastoma (GBM) is an aggressive primary malignant tumor of the adult central nervous system [1]. Standard care—maximal safe resection followed by radiotherapy and temozolomide—achieves a median overall survival of 14–16 months, and recurrence is nearly universal [2,3]. These outcomes reflect not only tumor-intrinsic heterogeneity but also a profoundly immunosuppressive tumor microenvironment (TME). Tumor-associated macrophages (TAMs) are the dominant immune population in GBM and display diverse functional states—proliferative, antigen-presenting, regulatory, interferon-responsive, hypoxic, and ribosome biogenesis programs—that collectively shape T-cell activity, metabolic tone, and vascular remodeling [4,5,6]. TAMs arise from two major sources—brain-resident microglia and infiltrating bone marrow-derived macrophages (BMDMs). These populations differ in localization and function: microglia (TMEM119^+^, P2RY12^+^) reside at the tumor margin, whereas BMDMs (CD45high, CD14^+^) accumulate in the hypoxic and perivascular core, where they adopt immunosuppressive, pro-angiogenic phenotypes. TAMs therefore exert both pro- and antitumor effects and exist along a continuum often described by the classical M1/M2 framework, with M1-like cells being inflammatory and M2-like cells supporting tissue repair and immune suppression. This heterogeneity highlights the need to identify molecular determinants that define TAM states in GBM [7]. However, the molecular determinants that mark these TAM states and link them to prognosis and therapy remain incompletely defined.

In the search for such determinants, the membrane-spanning 4-domains subfamily A (MS4A) gene cluster on chromosome 11, which encodes multi-pass transmembrane proteins involved in immune signaling, represents compelling candidates [8,9]. Beyond the well-known *MS4A1 (CD20)*, convergent evidence from neurodegeneration highlights membrane-spanning 4-domains A6A, *MS4A6A*,as a potential candidate gene in microglial biology: common variants at the MS4A locus associate with Alzheimer’s disease risk, with signals near *MS4A6A* and nominal brain eQTL effects [10,11]. *MS4A6A* is selectively enriched in microglia and increases with neuropathologic severity; in model systems, loss of the murine homolog *Ms4a6d* impairs plaque envelopment and phagocytosis and unleashes NF-κB-driven inflammation, whereas its overexpression dampens inflammatory signatures [12]. Beyond microglia, *MS4A6A* promotes endothelial dysfunction and monocyte adhesion via the IKK/NF-κB pathway in atherosclerosis [13]. Together, these data position *MS4A6A* as a key regulator of myeloid activation, phagocytic programs, and vascular inflammation—processes that map directly onto GBM’s TAM-rich, perivascular, and angiogenic niches.

Despite these compelling cross-disease insights, and although preliminary studies have linked *MS4A6A* to GBM prognosis, a comprehensive functional understanding is lacking [14,15]. The expression landscape, TAM-state specificity, spatial localization, and intercellular signaling consequences of *MS4A6A* in GBM have not been systematically defined. In this study, we integrate bulk- and single-cell RNA-seq to delineate the prognostic and expression profile of *MS4A6A*, map it across cellular lineages and TAM states, quantify *MS4A6A*-dependent myeloid–T-cell communication, and assess its spatial enrichment. We also use pharmacogenomics to nominate pathway vulnerabilities in an *MS4A6A*-defined risk context, ultimately positioning *MS4A6A* as a macrophage-anchored biomarker linking immune–stromal architecture to therapeutic opportunities in GBM.

## 2. Results

### 2.1. Identification of MS4A6A as an Immune-Related Prognostic Biomarker in GBM

Across the TCGA and two CGGA GBM cohorts, we first identified survival-associated genes (log-rank *p* < 0.05) and obtained 41 intersecting candidates (Figure 1A). Among these, *MS4A6A* emerged as an immune-related gene with consistent prognostic significance across datasets. Kaplan–Meier analyses showed that patients with high *MS4A6A* expression had significantly shorter overall survival (OS) in both CGGA (*p* = 0.037) and TCGA (*p* = 0.050) (Figure 1B,C). *MS4A6A* transcript levels were markedly elevated in GBM versus normal brain (*p* < 0.001; Figure 1D) and enriched in the mesenchymal molecular subtype (Figure 1E). Clinically, *MS4A6A* expression was higher in recurrent tumors and in older patients (>60 years) (Figure 1F,G), while no significant difference was observed for MGMT promoter methylation status (Figure 1H). Differential expression analysis between *MS4A6A*-high and *MS4A6A*-low tumors revealed upregulation of immune-activation programs, prominently MHC class II genes (e.g., HLA-DRA, HLA-DPA1, HLA-DPB1) and IL2RA, C1QB, alongside the downregulation of glycolytic/metabolic genes (ENO2, PFKP) and extracellular matrix components (VCAN, FN1) (Figure 1I). GSEA further indicated positive enrichment of inflammatory and immune-response signatures (ALLOGRAFT_REJECTION, IFN-γ_RESPONSE, IL6_JAK_STAT3_SIGNALING, TNFA_SIGNALING_VIA_NFKB) in the *MS4A6A*-high phenotype (Figure 1J).

### 2.2. MS4A6A Associates with Immune and Stromal Enrichment in the GBM Microenvironment

Bulk-level deconvolution demonstrated that *MS4A6A* expression positively correlates with ESTIMATE, stromal, and immune scores, indicating a more immune- and stroma-rich tumor milieu (Figure 2A–C). xCell profiling showed increased relative abundance of macrophages (both M1 and M2), dendritic cells, and multiple T-cell subsets in *MS4A6A*-high tumors (Figure 2D). These associations were corroborated by MCP-counter, which highlighted enrichment of myeloid lineages—monocytes and myeloid dendritic cells—in the *MS4A6A*-high group (Figure 2E). Consistently, ssGSEA revealed strong positive correlations between *MS4A6A* expression and macrophages, T follicular helper (Tfh) cells, and myeloid-derived suppressor cells (MDSCs) (r > 0.6, all *p* < 0.001; Figure 2F).

### 2.3. Single-Cell Atlas Localizes MS4A6A to TAMs and Reveals State-Specific Biological Programs

Integration of GBM single-cell RNA-seq defined major lineages, with selectively high *MS4A6A* expression in macrophages compared with malignant cells, T cells, and oligodendrocytes (Figure 3A,B). Within the macrophage compartment, we resolved functional TAM states—Regulatory-, Proliferation-, Transitory-, Ribo-, Phago/AP-, and IFN-TAMs (Figure 3C). *MS4A6A* displayed a broad, yet heterogeneous gradient across TAM states (Figure 3D), with *MS4A6A*-high cells most enriched in Ribo-TAM (203/230, 88.3%) and Regulatory-TAM (2417/3088, 78.3%), followed by Proliferation-(415/807, 51.4%), Transitory-(3375/7650, 44.1%), Phago/AP-(1393/3290, 42.3%), and IFN-TAMs (1041/2518, 41.3%) (Figure 3E).

Within each TAM state, *MS4A6A*-high cells exhibited coherent, state-specific programs (Figure 3F–K): Ribo-TAMs were enriched for RNA splicing, ribonucleoprotein complex biogenesis, and ribosome biogenesis; Regulatory-TAMs for leukocyte cell–cell adhesion, regulation of leukocyte activation, regulation of T cell activation, and antigen processing/presentation; Proliferation-TAMs for ribonucleoprotein and ribosome biogenesis, energy precursor generation, and chemotaxis; Transitory-TAMs for innate immune regulation, antiviral/type-I interferon responses, NF-κB signaling, and chemotaxis; Phago/AP-TAMs for innate immunity, immune effector regulation, energy metabolism, and leukocyte adhesion; and IFN-TAMs for regulation of innate immune response, vesicle transport/cell activation, and purine nucleotide metabolism. These data situate *MS4A6A* within immunoregulatory and interferon-linked TAM states that plausibly shape a highly inflamed, yet functionally suppressive GBM microenvironment. In contrast, *MS4A6A*-low TAMs (Appendix A) were enriched for migratory and homeostatic programs rather than immune or metabolic activation.

### 2.4. T-Cell Atlas and CellChat Highlight Strengthened Myeloid–T-Cell Communication in the MS4A6A-High Condition

After the annotation of six T-cell states (Activated, CD4, CD8, Proliferating, NK-like, and Regulatory/Treg; Figure 4A,B), TAMs were stratified by the median *MS4A6A* level (Figure 4C) and intercellular communication was quantified using CellChat [17]. Relative to the *MS4A6A*-low condition, the *MS4A6A*-high condition exhibited greater total numbers of interactions (3903 vs. 3005) and higher overall interaction strength (158.3 vs. 118.4) between TAMs and T cells (Figure 4D). Differential network analysis showed that increased edges were concentrated between *MS4A6A*-high TAM states and CD4 T cells and Treg (Figure 4E). Pathway weight heatmaps indicated Regulatory-, Ribo-, and Hypoxia-TAMs as major signal senders, with CD4 T cells and Treg as dominant receivers under the *MS4A6A*-high condition (Figure 4F). At the ligand–receptor level, upregulated pairs included PGE2–PTGES2–PTGER2/PTGER4, PECAM1–CD38, and THBS1–CD36, whereas FN1–(ITGA4/ITGB1) and LT4–LTC4S were reduced in the *MS4A6A*-high context (Figure 4G). Network centrality analyses highlighted: (Figure 4H) MPZ signaling across Regulatory-, Phago/AP-, Proliferation-, and Transitory-TAMs; (Figure 4I) broadly distributed prostaglandin (PGE_2_) signaling with Proliferating T cells, IFN- and Proliferation-TAMs as senders and CD4 T cells as primary receivers; (Figure 4J) PECAM1 signaling with high centrality in IFN- and Chemo-TAMs; and (Figure 4K) THBS (thrombospondin) signaling mainly sent by Chemo-TAMs to Proliferating T cells.

### 2.5. Spatial Enrichment in Vascular Niches and Nomination of Pathway-Targeted Vulnerabilities

Anatomical transcriptional profiling using the Ivy GAP dataset separated micro-dissected GBM regions by principal components (PC1 = 27.2%, PC2 = 17.3%; Figure 5A). *MS4A6A* expression was highest in angiogenic/perivascular compartments, including cellular tumor with microvascular proliferation (CT.mvp) and with hyperplastic blood vessels (CT.hbv) (Figure 5B). Drug-response modeling (OncoPredict) stratified by an *MS4A6A*-based risk group revealed higher predicted sensitivity in the High-Risk/*MS4A6A*-high group to AZD8055 (mTORC1/2 inhibitor), SCH772984 (ERK1/2 inhibitor), and Ribociclib (CDK4/6 inhibitor) (Figure 5C).

## 3. Discussion

GBM remains one of the most immunologically complex and treatment-refractory cancers. In this study, we identify *MS4A6A* as an immune-related prognostic biomarker that delineates a subset of tumor-associated macrophages enriched in inflammatory and immunoregulatory programs. Through bulk, single-cell, and spatial transcriptomic integration, we reveal that *MS4A6A*-high TAMs orchestrate an NF-κB–COX-2–PGE_2_ axis that strengthens myeloid–T-cell communication, particularly toward CD4^+^ T cells, within perivascular niches. These findings position *MS4A6A* as a macrophage-anchored regulator of prostaglandin-mediated immune suppression in GBM.

Our results demonstrate that *MS4A6A* is preferentially expressed in macrophage lineages, with pronounced enrichment in Regulatory- and Ribo-TAMs. This distribution suggests a link between *MS4A6A* and the transcriptional programs governing immune modulation and translational activity in macrophages. Previous studies in neurodegenerative disease contexts identified *MS4A6A* and its murine homolog Ms4a6d as microglia-specific genes that restrain inflammation and support phagocytic functions through NF-κB regulation [12]. Similarly, in atherosclerosis, *MS4A6A* promotes endothelial dysfunction and monocyte adhesion via the IKK/NF-κB pathway [13]. These reports converge with our observation that *MS4A6A* marks macrophage subsets with high inflammatory signaling yet tolerogenic potential, supporting its role as a myeloid signal integrator rather than a simple activation marker. The enrichment in Ribo-TAMs, a cluster defined by rRNA processing program and ribosomal genes (e.g., *RPL13*, *RPL19*), further implies that *MS4A6A*-high macrophages maintain enhanced biosynthetic and translational capacity. This is crucial as ribosome biogenesis is a key cellular process impaired by immunosuppressive PGE_2_-EP2/EP4 signaling in TME, and its preservation in these TAMs may provide the metabolic support for sustained cytokine and eicosanoid (including prostaglandin) production within the tumor microenvironment [18,19].

While bulk RNA-seq indicated a positive correlation between *MS4A6A* and MHC class II gene expression (Figure 1I), our single-cell resolution analysis clarifies that this association is not primarily driven by their co-expression within the same cellular state. Instead, *MS4A6A* and classical MHC II genes are partitioned across different TAM subsets—with *MS4A6A* marking Regulatory-, Ribo-, and Proliferation-TAMs, and MHC II defining the Phago/AP-TAM compartment. Given that *MS4A6A*-high tumors exhibit expanded myeloid infiltration, the bulk signature likely captures a microenvironment where these distinct, specialized macrophage populations coexist. The correlation reflects a coordinated system shift toward an immune-rich, pro-presenting niche, not a coregulated transcriptional module. This finding highlights the critical need to complement bulk omics with single-cell approaches to accurately interpret co-expression patterns in complex tissues.

Our CellChat analysis suggests the prostaglandin pathway as one of the most prominent upregulated communication networks in the *MS4A6A*-high condition. PGE_2_ is a lipid mediator synthesized by cyclooxygenase-2 (COX-2/PTGS2) and prostaglandin E synthases, known to exert profound effects on antitumor immunity [20]. In multiple cancers, PGE_2_ suppresses CD4^+^ and CD8^+^ T-cell proliferation, downregulates IFN-γ and IL-2 production, and fosters the expansion of regulatory or IL-10–producing Tr1-like cells through EP2/EP4-cAMP signaling. Mechanistically, as revealed by single-cell studies, PGE_2_ signaling upon T cell activation downregulates IL-2 receptor alpha (IL2RA), impairing IL-2-STAT5 signaling. This leads to suppression of c-Myc and PGC-1α, resulting in coordinated downregulation of oxidative phosphorylation, glycolysis, and ribosomal biogenesis, ultimately crippling T cell expansion, survival, and antitumor function [21]. Consistent with this paradigm, our dataset demonstrates that TAMs—particularly IFN- and Proliferation-TAMs—act as dominant senders and influencers in the PGE_2_ network, while CD4^+^ T cells serve as the principal receivers. This cell-resolved evidence provides a concrete map of how PGE_2_ signaling is organized in GBM: a myeloid-centered loop where TAM-derived prostaglandins modulate CD4^+^ T-cell behavior, establishing an “inflamed but ineffective” immune microenvironment. Importantly, *MS4A6A*-high tumors also showed positive enrichment of TNF-α/NF-κB and IL6/JAK/STAT3 signatures, two upstream activators of COX-2, suggesting that *MS4A6A* may operate within a self-reinforcing inflammatory circuit that sustains prostaglandin output [22]. This circuit mirrors the spatial niche-driven reprogramming observed in other TAM subsets, such as Hypoxia-TAMs, which are steered by local cues to adopt a distinct, vasculature-disrupting phenotype, underscoring the broader principle that TAM functional diversity is critically shaped by specific TME niches.

Spatial transcriptomic data from the Ivy GAP dataset localized *MS4A6A* to the angiogenic and perivascular compartments, which are established immune regulatory niches in GBM. Perivascular TAMs have been shown to shape endothelial activation, promote leukocyte adhesion, and orchestrate the trafficking of peripheral T cells into the tumor [23]. The co-localization of *MS4A6A* with vascular and adhesion molecules (PECAM1, THBS1) indicates that *MS4A6A*-high TAMs may represent a perivascular subpopulation specialized in modulating immune cell entry and functional reprogramming. Within this niche, high prostaglandin signaling could transiently attract CD4^+^ T cells while simultaneously suppressing their effector differentiation, thereby reinforcing local immune tolerance. This interpretation aligns with prior studies demonstrating that endothelial-associated macrophages produce PGE_2_ and VEGF to maintain an immunosuppressive angiogenic loop [24].

The functional prominence of this prostaglandin-driven immunosuppressive circuit is further underscored by its therapeutic vulnerability. Our pharmacogenomic analysis revealed that *MS4A6A*-high GBM samples displayed greater predicted sensitivity to mTORC1/2 and ERK1/2 inhibitors (AZD8055, SCH772984). This is highly relevant as these pathways are known to converge on inflammatory lipid metabolism and COX-2 expression: mTOR signaling promotes prostaglandin synthesis in macrophages by increasing PTGS2 translation and lipid droplet formation [25,26], whereas ERK activation enhances downstream EP receptor signaling [27]. Therefore, the observed drug sensitivity suggests that a self-reinforcing mTOR–ERK–COX-2 axis may sustain the prostaglandin program in perivascular, *MS4A6A*-high TAMs, and its disruption could simultaneously attenuate both inflammation and immune suppression. Moreover, preclinical studies have shown that directly targeting this pathway via EP4 antagonists or COX-2 inhibitors can restore antitumor T-cell responses and synergize with immune checkpoint blockade [28]. These findings collectively underscore the potential of *MS4A6A* as a spatial biomarker to stratify patients who might benefit from combinatorial regimens that simultaneously target the PGE_2_/EP4 signaling axis and its upstream mTOR/ERK drivers.

Several limitations should be acknowledged. First, our analyses are correlative and based on transcriptomic inference; functional experiments (e.g., *MS4A6A* knockdown in macrophage–T-cell co-culture or organoid models) are needed to confirm its causal role in regulating the NF-κB–COX-2–PGE_2_ axis and, consequently, the ribosome biogenesis and bioenergetic programs in TAMs. Second, while we identified enhanced PGE_2_ signaling, its direct impact on T-cell function—such as the suppression of IL-2 signaling, downregulation of c-Myc/PGC-1α, and the resultant impairment in oxidative phosphorylation and glycolysis as reported in other cancers—remains to be experimentally validated in GBM. Third, the clinical cohorts analyzed are retrospective; prospective studies with immunotherapy-treated GBM patients will be valuable to test the predictive power of *MS4A6A*. In addition, the predicted drug sensitivity differences derived from OncoPredict are hypothesis-generating and will require further validation through pharmacological assays in experimental models. Finally, although our data primarily support a PGE_2_-centered mechanism, the upregulation of THBS1 and NF-κB activity suggests a possible link between this pathway and “don’t-eat-me” checkpoints (e.g., CD47–SIRPα, CD24–SIGLEC10) that merits further investigation.

## 4. Materials and Methods

### 4.1. Data Acquisition and Preprocessing

Transcriptomic and clinical data for GBM were obtained from The Cancer Genome Atlas (TCGA, https://portal.gdc.cancer.gov/) and the Chinese Glioma Genome Atlas (CGGA, http://www.cgga.org.cn/) [29]. Raw counts data were downloaded (10 March 2025) and normalized using DESeq2 in R. Samples with incomplete survival data were excluded. Differential expression between GBM and normal brains for *MS4A6A* was examined via GEPIA (http://gepia.cancer-pku.cn/). To characterize cell type-specific expression of *MS4A6A*, single-cell RNA sequencing (scRNA-seq) data from newly diagnosed GBM were obtained from the Brain Immune Atlas (https://www.brainimmuneatlas.org/) [30]. scRNA-seq data were processed with Seurat [31]. Cells were filtered based on the following criteria: number of unique genes detected (nFeature_RNA) between 200 and 6000, percentage of mitochondrial reads less than 15%, to remove low-quality cells and potential doublets. Major lineages (malignant cells, tumor-associated macrophages, T cells, oligodendrocytes) were identified using canonical markers. Regionally annotated bulk RNA-seq profiles from the Ivy Glioblastoma Atlas Project (Ivy GAP; http://glioblastoma.alleninstitute.org/) were used to evaluate the spatial distribution of *MS4A6A* across anatomic compartments (e.g., CT, CT.mvp, CT.hbv, CT.pnz, IT, LE) [23].

### 4.2. Differential Expression and Survival Analysis of MS4A6A

Patients in TCGA and CGGA were dichotomized into *MS4A6A*-high and *MS4A6A*-low groups by median expression. Kaplan–Meier and univariable Cox analyses were performed using survival/survminer to compare overall survival (OS) between groups. For bulk differential expression, normalized RNA-seq data were analyzed using limma (version 3.64.3) [32] with empirical Bayes moderation; significantly differentially expressed genes (DEGs) were defined as adjusted *p* < 0.05 and |log2 fold change| > 1. Volcano plots were generated with ggplot2/ggrepel.

### 4.3. Functional Enrichment, Immune Infiltration, and GSEA/Correlation Analyses

DEGs identified by limma were subjected to Gene Ontology (GO) and KEGG pathway enrichment using clusterProfiler and org.Hs.eg.db [33]. Gene Set Enrichment Analysis (GSEA) [34] was performed on a ranked gene list (moderated t-statistic) with fgsea/enrichplot to evaluate Hallmark pathways. Multiple testing was controlled using the Benjamini–Hochberg false discovery rate (FDR). To quantify tumor microenvironment features, immune and stromal components were estimated from bulk RNA-seq using MCP-counter [35] and xCell [36], while tumor purity, stromal, and immune scores were computed with ESTIMATE [37]. Heatmaps summarized cell-type abundance patterns across groups. To further profile immune activity, ssGSEA [38] was applied to curated immune gene sets; enrichment scores were correlated with *MS4A6A* expression using Pearson (or Spearman where appropriate).

### 4.4. Single-Cell RNA-Seq Analysis and Cell-Type Annotation

scRNA-seq data were processed using Seurat (v5). After standard QC, data were log-normalized; 2000 highly variable genes were identified by the “vst” method implemented in Seurat. Following scaling and Principal Component Analysis (PCA), graph-based clustering used the first 20 PCs at resolution 0.25. UMAP was used for 2D visualization. Cluster annotation followed a multi-step strategy: (i) canonical markers for broad lineages; (ii) FindAllMarkers (Wilcoxon; thresholds: log2FC > 0.25, expressed in >25% of cells) to identify cluster-specific markers; (iii) enrichment-based functional interpretation (clusterProfiler/org.Hs.eg.db). Sub-clustering of the macrophage compartment identified eight distinct TAM functional states—Proliferation-TAM, IFN-TAM, Regulatory-TAM, Phago/AP-TAM, Hypoxia-TAM, Chemo-TAM, Ribo-TAM, and Transitory-TAM [16]. Similarly, sub-clustering of the T cell compartment resolved six distinct T cell states: Activated T cells, CD4^+^ T cells, CD8^+^ T cells, Proliferating T cells, NK-like T cells, and Regulatory T cells (Tregs), based on the expression of canonical lineage and functional markers [39,40].

### 4.5. Single-Cell Resolution Analysis of MS4A6A in TAM Subpopulations

To delineate the functional role of *MS4A6A* across TAM states, we performed a state-specific analysis within the eight identified TAM subpopulations. For each state, cells were stratified into *MS4A6A*-High and *MS4A6A*-Low groups based on the median expression level of *MS4A6A* within that specific state. Differential gene expression analysis between these paired groups was conducted using Seurat’s FindMarkers function (Wilcoxon rank-sum test; logfc.threshold = 0.25, min.pct = 0.1). Genes meeting the thresholds of |avg_log2FC| > 0.25 and a Benjamini–Hochberg adjusted *p*-value < 0.05 were considered significant. Gene Ontology Biological Process enrichment analysis was subsequently performed on the significant differentially expressed genes for each TAM state using the enrichGO function from the clusterProfiler package (pvalueCutoff = 0.05, qvalueCutoff = 0.2, ont = “BP”), set.seed(12345) was used for reproducibility.

### 4.6. Cell–Cell Communication Analysis

Intercellular signaling was inferred using CellChat (v1.6.0) [17]. TAMs were stratified into *MS4A6A*-High and *MS4A6A*-Low groups (median split in TAMs), and separate CellChat objects were constructed per group. Communication probability and pathway-level information flow were computed via computeCommunProb/computeCommunProbPathway and aggregated with aggregateNet. Groupwise comparisons used mergeCellChat and rankNet to quantify differences in total number and overall strength of interactions. Network centrality analyses (netAnalysis) defined sender/receiver/mediator/influencer roles. Differential interaction networks were visualized with circle plots and pathway heatmaps, and ligand–receptor (L–R) contrasts were summarized in volcano plots.

### 4.7. Spatial Transcriptomic Localization and Drug Sensitivity Prediction

The spatial expression pattern of *MS4A6A* was assessed using regionally micro-dissected glioblastoma transcriptomes from the Ivy GAP database [23]. PCA was employed to distinguish the anatomic compartments. Differential *MS4A6A* expression across these regions was evaluated. To investigate therapeutic implications, we predicted drug sensitivity using the OncoPredict algorithm (v1.2.0) [41]. GBM bulk transcriptomic profiles served as input, with drug-response models trained on CTRP pharmacogenomic datasets [42]; batch effects were corrected by ComBat [43]. Lower predicted Area Under the Curve (AUC) indicates higher sensitivity. Group differences were tested by Wilcoxon rank-sum with Benjamini–Hochberg (BH) correction across compounds.

### 4.8. Statistical Analysis

Analyses were performed in R (v4.3.0). Two-sided *p* < 0.05 was considered significant unless specified. For bulk comparisons, Wilcoxon rank-sum or Student’s *t*-test was used as appropriate. Survival analyses used Kaplan–Meier (log-rank) and Cox proportional hazards regression. Correlations used Pearson or Spearman with BH FDR correction for multiple testing. For single-cell DE, Wilcoxon tests with BH adjustment were used as implemented in Seurat. Enrichment analyses controlled FDR via BH. Plotting used ggplot2 (v3.5.x), viridis, and Seurat’s built-in visualization functions.

## 5. Conclusions

In summary, this study defines *MS4A6A* as a macrophage-anchored regulator enriched in Ribo-TAMs, which shapes an immunosuppressive, perivascular niche in GBM by activating the NF-κB–COX-2–PGE_2_ signaling axis. *MS4A6A*-high TAMs act as central hubs transmitting prostaglandin signals to CD4^+^ T cells, reprogramming them into a state of functional impairment, potentially through the suppression of bioenergetic and translational capacity. These findings bridge genetic and microenvironmental mechanisms of immune evasion and highlight *MS4A6A* both as a mechanistic biomarker identifying a therapeutically vulnerable TAM subpopulation and as a potential guide for rational combinations of PGE_2_/EP4 and ERK/mTOR-targeted therapies in GBM.

## Figures and Tables

**Figure 1 ijms-27-00058-f001:**
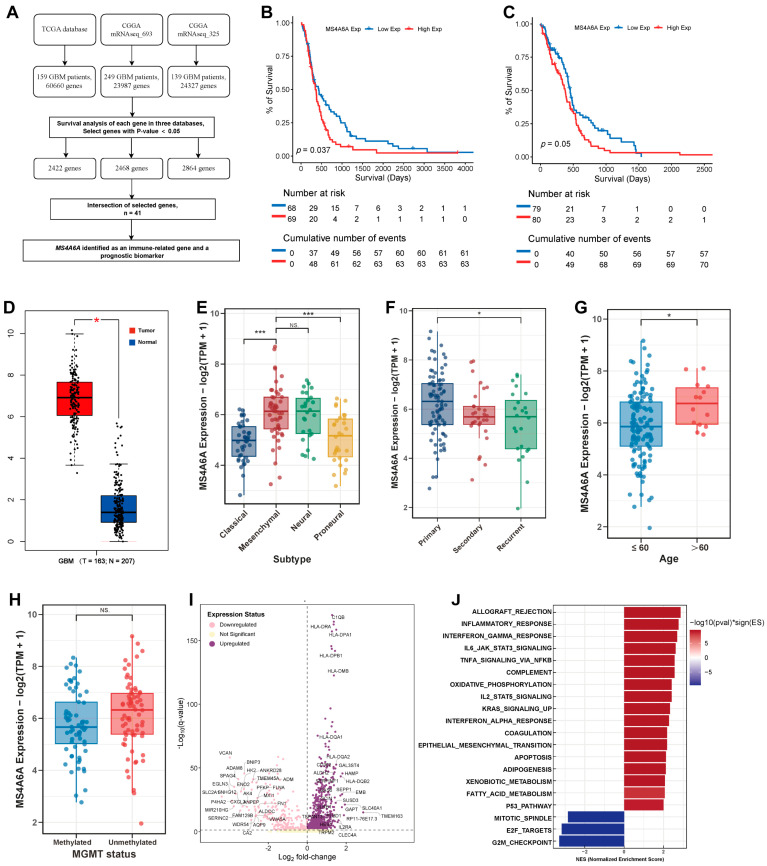
Identification and characterization of MS4A6A as an immune-related prognostic biomarker in GBM. (**A**) Workflow of gene selection and validation. Survival-associated genes were first identified from the TCGA and two CGGA GBM cohorts (cutoff *p* < 0.05), and their intersection yielded 41 candidates. Among them, *MS4A6A* was recognized as an immune-related gene with consistent prognostic significance across datasets. (**B**,**C**) Kaplan–Meier survival analyses in the CGGA (**B**) and TCGA (**C**) cohorts demonstrated that patients with high *MS4A6A* expression exhibited significantly shorter overall survival compared with the low-expression group (*p* = 0.037 and *p* = 0.05, respectively). (**D**) Boxplot showing elevated *MS4A6A* expression in GBM tumors relative to normal brain tissues (*p* < 0.001). (**E**) Expression levels of *MS4A6A* across molecular subtypes revealed enrichment in the mesenchymal subtype. (**F**,**G**) Clinical correlation analyses showed higher *MS4A6A* expression in recurrent tumors (**F**) and in older patients (>60 years) (**G**). (**H**) No significant difference in *MS4A6A* expression was observed between MGMT-methylated and unmethylated tumors. (**I**) The volcano plot displays DEGs between the *MS4A6A*-high and *MS4A6A*-low groups. A significant upregulation was observed for genes involved in immune activation, including a broad set of Major Histocompatibility Complex (MHC) class II genes (e.g., *HLA-DRA*, *HLA-DPA1*, *HLA-DPB1*) and other immune activators like *IL2RA* and *C1QB*. Conversely, the downregulation of genes related to cellular metabolism (e.g., glycolytic enzymes *ENO2*, *PFKP*) and microenvironment interaction (e.g., extracellular matrix components *VCAN*, *FN1*) suggests a functional shift. (**J**) Gene Set Enrichment Analysis (GSEA) revealed that the *MS4A6A*-high phenotype was positively associated with inflammatory and immune-response signatures—including ALLOGRAFT_REJECTION, IFN-γ_RESPONSE, IL6_JAK_STAT3_SIGNALING, and TNFA_SIGNALING_VIA_NFKB—and negatively associated with proliferation-related hallmarks (E2F_TARGETS, G2M_CHECKPOINT). Statistical significance is denoted as follows: *p* < 0.05 (*), and *p* < 0.001 (***); NS indicates no statistical significance.

**Figure 2 ijms-27-00058-f002:**
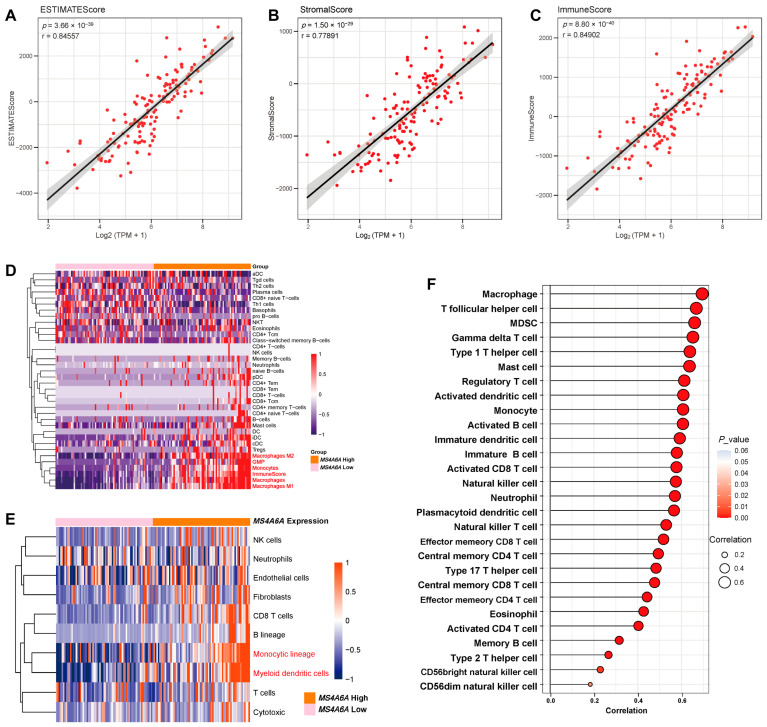
*MS4A6A* expression is strongly associated with immune and stromal enrichment in the GBM microenvironment. (**A**–**C**) Correlation analysis between *MS4A6A* expression and three scores derived from the ESTIMATE algorithm: (**A**) ESTIMATE score, (**B**) stromal score, and (**C**) immune score. Higher *MS4A6A* expression was significantly associated with elevated stromal and immune enrichment (n = 159). (**D**) Heatmap generated by the xCell algorithm showing the relative abundance of diverse immune cell types between *MS4A6A*-high and *MS4A6A*-low groups. Samples with high MS4A6A expression displayed enhanced infiltration of macrophages (particularly M1 and M2 phenotypes), dendritic cells, and multiple T-cell subsets (n = 159). (**E**) MCP counter-analysis further validated the association between *MS4A6A* and the enrichment of myeloid lineages—including monocytes, macrophages, and myeloid dendritic cells—consistent with an inflammatory and stromal-rich tumor niche (n = 159). (**F**) Bubble plot showing the correlations between *MS4A6A* expression and the relative abundance of 24 immune cell populations estimated by ssGSEA. Macrophages, T follicular helper cells, and myeloid-derived suppressor cells (MDSCs) exhibited the strongest positive correlations (*r* > 0.6, *p* < 0.001) (n = 159).

**Figure 3 ijms-27-00058-f003:**
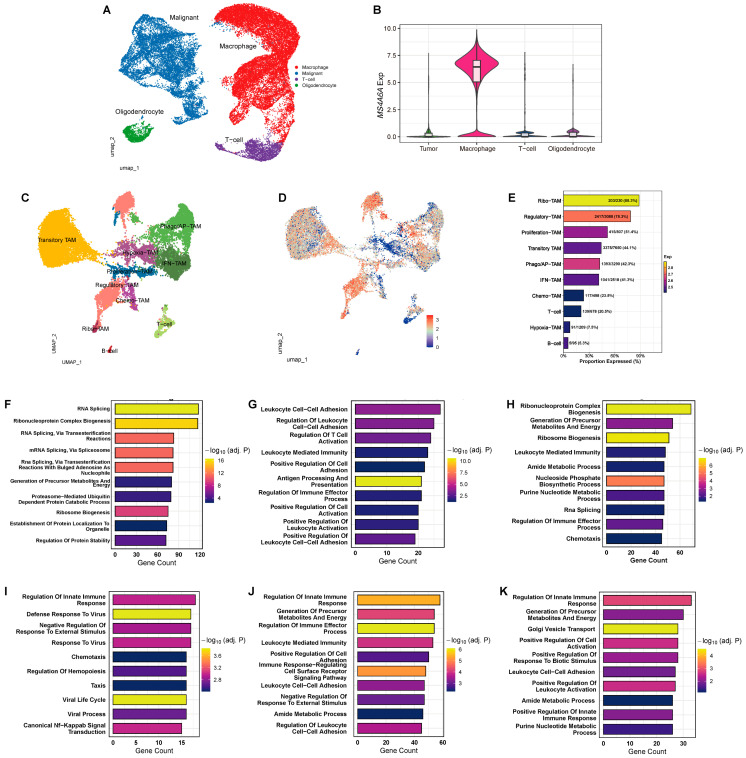
Single-cell atlas locates *MS4A6A* to TAMs and reveals subtype-specific biological programs. (**A**) UMAP of the integrated GBM scRNA-seq dataset showing the major lineages (malignant cells, macrophages, T cells, oligodendrocytes). (**B**) Violin plot of *MS4A6A* across lineages demonstrates selectively high expression in macrophages. (**C**) UMAP of the tumor-associated macrophage compartment resolved into functional states: Regulatory-TAM, Proliferation-TAM, Transitory-TAM, Ribo-TAM, Phago/AP-TAM (phagocytosis/antigen presentation), and IFN-TAM. Reproduced from ref. [16], *Cancers* 2025, *17*(19), 3271; https://doi.org/10.3390/cancers17193271 (Figure 3E). (**D**) Feature plot of *MS4A6A* within TAMs showing a broad yet heterogeneous gradient of expression across states. (**E**) Proportion and average level of *MS4A6A* expression across clusters. *MS4A6A*-high cells were most enriched in Ribo-TAM (203/230, 88.3%) and Regulatory-TAM (2417/3088, 78.3%), followed by Proliferation-TAM (415/807, 51.4%), Transitory-TAM (3375/7650, 44.1%), Phago/AP-TAM (1393/3290, 42.3%), and IFN-TAM (1041/2518, 41.3%). (**F**–**K**) Gene Ontology (GO, Biological Process) enrichment based on differentially expressed genes between *MS4A6A*-high and *MS4A6A*-low cells within each TAM state. (**F**) Ribo-TAM: enrichment for RNA splicing, ribonucleoprotein complex biogenesis, rRNA/ncRNA processing, translation/ribosome biogenesis, and protein targeting. (**G**) Regulatory-TAM: enrichment for leukocyte cell–cell adhesion, regulation of leukocyte activation, antigen processing and presentation, positive regulation of cytokine production, and leukocyte-mediated immunity. (**H**) Proliferation-TAM: enrichment for ribonucleoprotein complex biogenesis, generation of precursor metabolites and energy, nucleoside/nucleotide metabolic processes, ribosome biogenesis, and chemotaxis. (**I**) Transitory-TAM: enrichment for regulation of innate immune response, defense response to virus, type-I interferon response, negative regulation of viral genome replication, chemotaxis/taxis, and canonical NF-κB signaling. (**J**) Phago/AP-TAM: enrichment for regulation of innate immune response, generation of precursor metabolites and energy, response to interferon-β/type-I interferon, leukocyte-mediated immunity, antigen processing/presentation, and regulation of leukocyte cell–cell adhesion. (**K**) IFN-TAM: enrichment for innate antiviral programs, including type-I interferon and interferon-β responses, regulation of viral processes, leukocyte-mediated immunity, and purine nucleotide metabolism.

**Figure 4 ijms-27-00058-f004:**
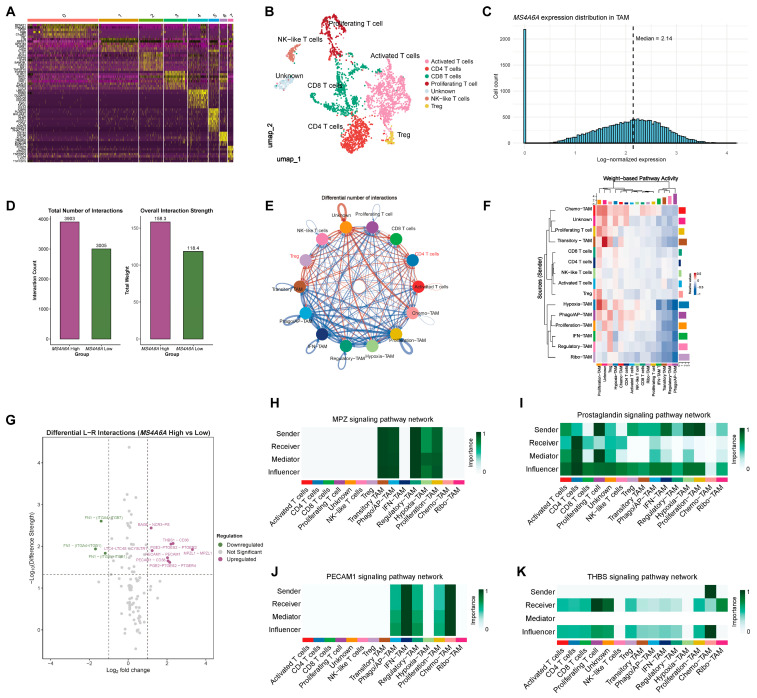
T-cell atlas and CellChat comparison of *MS4A6A*-high vs. *MS4A6A*-low TAMs reveal strengthened myeloid–T-cell communications. (**A**) Heatmap of canonical marker genes used to define major T-cell states in the single-cell RNA-seq dataset. (**B**) UMAP visualization showing seven T-cell states, including Activated, CD4, CD8, Proliferating, NK-like, and Regulatory (Treg) cells. (**C**) Distribution of *MS4A6A* expression in TAMs. The dashed line marks the median expression level used to stratify TAMs into *MS4A6A*-high and *MS4A6A*-low groups. (**D**) Summary of CellChat analysis comparing interactions between T cells and TAMs under *MS4A6A*-high and *MS4A6A*-low conditions. Both the total number of interactions (3903 vs. 3005) and overall interaction strength (158.3 vs. 118.4) were higher in the *MS4A6A*-high group. (**E**) Differential interaction network (circle plot) between T-cell states and TAM states comparing *MS4A6A*-high vs. *MS4A6A*-low groups. Red and blue edges denote interactions increased (red) or decreased (blue) in the *MS4A6A*-high group, respectively. The increases are concentrated between *MS4A6A*-high TAM states and CD4 T cells and Treg. (**F**) Heatmap of weight-based pathway activity illustrating the relative strength of ligand–receptor signaling between TAM subtypes and T-cell populations in the *MS4A6A*-high condition. Warmer colors represent higher signaling weights. Regulatory-, Phago/AP-, and IFN-TAMs emerge as major signal senders, whereas CD4 T cells and Treg act as dominant receivers. (**G**) Volcano plot showing differential ligand–receptor (L–R) interactions between *MS4A6A*-high and *MS4A6A*-low TAM groups. Notably upregulated interactions include PGE2–PTGES2–PTGER2/PTGER4, PECAM1–CD38, and THBS1–CD36, whereas downregulated pairs such as FN1–(ITGA4/ITGB1) and LT4–LTC4S are enriched in the *MS4A6A*-low condition. (**H**–**K**) Network centrality analyses highlighting key signaling pathways upregulated in *MS4A6A*-high TAM–T-cell communication. (**H**) MPZ signaling exhibits strong interactions in Regulatory-, Phago/AP-, Proliferation-, and Transitory-TAMs. (**I**) Prostaglandin (PGE_2_) signaling is broadly distributed, with Proliferating T cells, IFN-, Proliferation-TAMs serving as major senders, and CD4 T cells as primary receivers. (**J**) PECAM1 signaling displays high centrality in IFN-TAMs and Chemo-TAMs. (**K**) THBS (Thrombospondin) signaling is mainly mediated by Chemo-TAMs as senders and Proliferating T cells as receivers.

**Figure 5 ijms-27-00058-f005:**
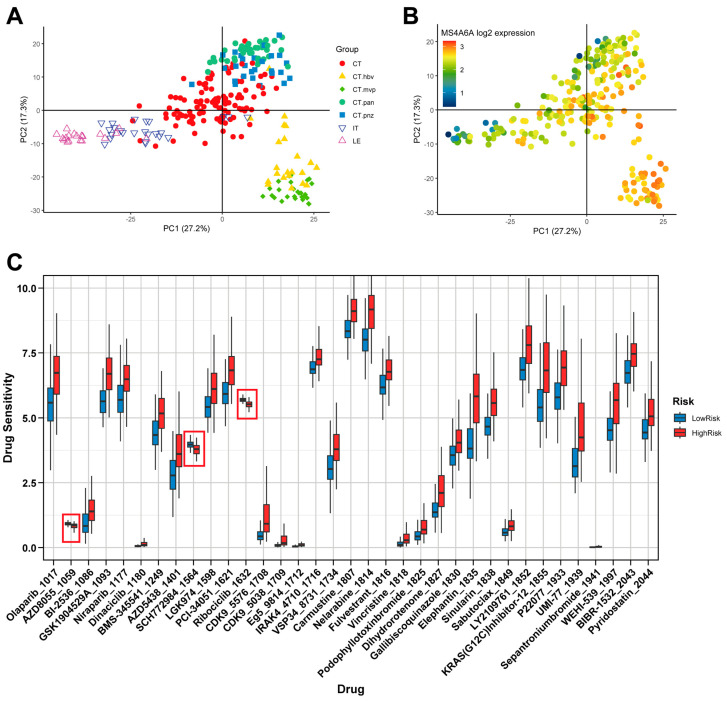
Ivy GAP anatomy shows *MS4A6A* enrichment in vascular niches of GBM and nominates pathway-targeted vulnerabilities. (**A**) Principal Component Analysis (PCA) of Ivy GAP micro-dissected GBM regions (PC1 = 27.2%, PC2 = 17.3%). Samples are annotated by anatomic compartment: CT (Cellular Tumor), CT.hbv (CT—Hyperplastic Blood Vessels), CT.mvp (CT—Microvascular Proliferation), CT.pan (CT—Pseudopalisading cells around Necrosis), CT.pnz (CT—Peri-necrotic Zone), IT (Infiltrating Tumor), and LE (Leading Edge). Reproduced from ref. [16], *Cancers* 2025, 17(19), 3271; https://doi.org/10.3390/cancers17193271 (Figure 5F). (**B**) *MS4A6A* is highly expressed in CT.mvp and CT.hbv, which indicates that *MS4A6A* preferentially marks angiogenic/perivascular microenvironments in GBM. (**C**) Predicted drug sensitivity profiles (OncoPredict) for patients stratified by the *MS4A6A*-based risk group (HighRisk = *MS4A6A*-high; LowRisk = *MS4A6A*-low). The y-axis represents the predicted IC50, with lower values indicating greater sensitivity. The HighRisk group shows significantly lower predicted IC50 for AZD8055 (mTORC1/2 inhibitor), SCH772984 (ERK1/2 inhibitor), and Ribociclib (CDK4/6 inhibitor), highlighted with boxed annotations.

## Data Availability

The data supporting the findings of this study are available upon reasonable request from the corresponding author. The R code used in this study will be made available at GitHub (https://github.com/cielowq?tab=repositories, accessed on 17 December 2025).

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
