# Peer review of "The Myeloid Biomarker MS4A6A Drives an Immunosuppressive Microenvironment in Glioblastoma via Activation of the PGE2 Signaling Axis"

_ijms, 2025, doi:10.3390/ijms27010058_

Round 1
Reviewer 1 Report
Comments and Suggestions for Authors
This manuscript analyses available datasets to produce bioinformatics data and determine connections of the MS4A6A gene to immune-regulation in the glioblastoma. The authors work contain some significant findings. However, additional analysis would conclusively provide key evidence for their overall outcomes of this manuscript. I have specific recommendations outlined below.
- The introduction is limited. The manuscript has a strong focus on macrophages. However the discussion on infiltrating macrophages and microglia is sparse. Likewise, discussion on the different TAMs studies in this manuscript should also be added to the introduction, highlighting their pro- or anti-tumor properties. Similarly, M1 and M2 macrophages should be explained.
- Which markers are used to distinguish M1 from M2 macrophages in Fig 2? And which subgroup express more MS4A6A?
- Likewise, which gene expression markers are used to distinguish the different TAMs from fig 3?
- Were the MHC II genes found to correlate with high MS4A6A gene from the bulk screening also shown to correlate with MS4A6A expression in the specific macrophage (and T cell) populations? The authors discuss this...especially if they do not.
Author Response
Response to Reviewer 1
This manuscript analyses available datasets to produce bioinformatics data and determine connections of the MS4A6A gene to immune-regulation in the glioblastoma. The authors work contain some significant findings. However, additional analysis would conclusively provide key evidence for their overall outcomes of this manuscript. I have specific recommendations outlined below.
- The introduction is limited. The manuscript has a strong focus on macrophages. However the discussion on infiltrating macrophages and microglia is sparse. Likewise, discussion on the different TAMs studies in this manuscript should also be added to the introduction, highlighting their pro- or anti-tumor properties. Similarly, M1 and M2 macrophages should be explained.
Response:
We thank the reviewer for this helpful suggestion. In the revised manuscript, we have expanded the Introduction to provide a clearer immunologic context for our work. We added below paragraph to the introduction part:
“TAMs arise from two major sources—brain-resident microglia and infiltrating bone marrow–derived macrophages (BMDMs). These populations differ in localization and function: microglia (TMEM119⁺, P2RY12⁺) reside at the tumor margin, whereas BMDMs (CD45high, CD14⁺) accumulate in the hypoxic and perivascular core where they adopt immunosuppressive, pro-angiogenic phenotypes. TAMs therefore exert both pro- and anti-tumor effects and exist along a continuum often described by the classical M1/M2 framework, with M1-like cells being inflammatory and M2-like cells supporting tissue repair and immune suppression. This heterogeneity highlights the need to identify molecular determinants that define TAM states in GBM[7].”
- Which markers are used to distinguish M1 from M2 macrophages in Fig 2? And which subgroup express more MS4A6A?
Response:
Thank you for your comment. In Fig. 2, the M1 and M2 macrophage signatures were defined according to the standard gene sets implemented in the xCell and MCP-counter algorithm. M1 macrophages are characterized by inflammatory markers such as IL1B, IL6, TNF, CXCL9, and CXCL10; M2 macrophages are defined by immunoregulatory markers including CD163, MRC1 (CD206), MSR1, and IL10. Regarding MS4A6A expression, the heatmaps display samples ordered from low to high MS4A6A expression. The MS4A6A-high group shows a pronounced enrichment of myeloid lineages, particularly M2-like macrophages, followed by M1 macrophages and myeloid dendritic cells. Thus, the M2 macrophage subgroup is most strongly associated with high MS4A6A expression.
- Likewise, which gene expression markers are used to distinguish the different TAMs from fig 3?
Response:
Thank you for this question. Annotation of TAM states in Fig. 3 followed a three step procedure: (1) canonical macrophage lineage markers; (2) cluster-specific marker genes identified by FindAllMarkers; and (3) functional enrichment analysis for biological interpretation. Using these strategy, eight TAM functional states were defined. Specifically, representative gene markers for each subtype are listed below: Proliferation-TAM: enriched for cell-cycle genes such as MKI67, TOP2A, PCNA, and BIRC5; IFN-TAM: including IFIT1, IFIT3, ISG15, CXCL9, and STAT1; Regulatory-TAM: expressing immunoregulatory genes such as CD163, MRC1 (CD206), IL10, and MSR1; Phago/AP-TAM: including HLA-DRA, HLA-DRB1, CD74, and CTSS; Hypoxia-TAM: HIF1A, VEGFA, ADM, and SLC2A1; Chemo-TAM CCL3, CCL4, CCL3L1; Ribo-TAM: RPL and RPS; Transitory-TAM: according to the GO results, representing a transitional state between the above phenotypes. These representative markers were derived from cluster-specific differential expression results combined with functional enrichment, consistent with established TAM classifications in glioblastoma.
- Were the MHC II genes found to correlate with high MS4A6A gene from the bulk screening also shown to correlate with MS4A6A expression in the specific macrophage (and T cell) populations? The authors discuss this...especially if they do not.
Response:
We sincerely thank the reviewer for this expert and insightful comment. As noted, the bulk RNA-seq analysis showed that high MS4A6A expression was accompanied by elevated levels of multiple MHC II genes. Within the macrophage atlas, we found that MS4A6A expression was most enriched in Regulatory-, Ribo- and Proliferation-TAM states, whereas classical MHC II genes were predominantly elevated in the Phago/AP-TAM population. The strong association between MS4A6A and MHC II observed in bulk RNA-seq likely arises not from obligate co-expression within individual macrophage states, but rather from compositional shifts in the tumor microenvironment—namely, that MS4A6A-high tumors harbor greater myeloid infiltration, including antigen-presenting Phago/AP-TAMs, which collectively amplify the MHC II signal. We have added a explanation to the Discussion to clarify this important distinction between cell-intrinsic and compositional effects.
We added this to the discussion part of the manuscript ” While bulk RNA-seq indicated a positive correlation between MS4A6A and MHC class II gene expression (Fig. 1I), our single-cell resolution analysis clarifies that this association is not primarily driven by their co-expression within the same cellular state. Instead, MS4A6A and classical MHC II genes are partitioned across different TAM subsets—with MS4A6A marking Regulatory-, Ribo-, and Proliferation-TAMs, and MHC II defining the Phago/AP-TAM compartment. Given that MS4A6A-high tumors exhibit expanded myeloid infiltration, the bulk signature likely captures a microenvironment where these distinct, specialized macrophage populations coexist. The correlation reflects a coordinated system shift toward an immune-rich, pro-presenting niche, not a coregulated transcriptional module. This finding highlights the critical need to complement bulk omics with single-cell approaches to accurately interpret co-expression patterns in complex tissues.”

Reviewer 2 Report
Comments and Suggestions for Authors
Overall this manuscript is in goof shape. However, a few issues need to be addressed.
First, for overall experimental design, is there any reason why author choose to use median as cutoff, instead of more strict cutoffs, such as top/bottom quantiles?
There are some comments for the results section:
For figure 1A, the author should try to explain the reason why from three dataset, there are only a very small fraction of overlapping genes. Also, it would be better to provide a visualization to show the top genes and more reasonales in the context to explain why choose MS4A6A.
For figure 2, author can clarify the origin of the data used in the plot. Also, for correlation plots, author can add the number of patients. For heatmaps (figure 2D-E), it would be nice to include the expression of MS4A6A as a part of annotation. Also, the color code for two heatmap should be consistent.
For Figure 5C, the compounds mentioned in cnotext in general have a lower drug sensitivity and difference between high and low groups is minimal. Author should justify why the small difference could make biological difference in clinical application. Also the plot needs to be rearranged or highlight the selected terms to make it easy to read.
Author Response
Response to Reviewer 2
Overall this manuscript is in good shape. However, a few issues need to be addressed.
- First, for overall experimental design, is there any reason why author choose to use median as cutoff, instead of more strict cutoffs, such as top/bottom quantiles?
Response:
We thank the reviewer for this question. We chose the median cutoff primarily to maintain statistical power and ensure reproducibility across our validation cohorts. Median splits yield balanced groups, which is critical in modest-sized GBM datasets. Also, this method consistent with common practice in similar transcriptomic biomarker studies.
- For figure 1A, the author should try to explain the reason why from three dataset, there are only a very small fraction of overlapping genes. Also, it would be better to provide a visualization to show the top genes and more reasonales in the context to explain why choose MS4A6A.
Response:
We thank the reviewer for this comment. The limited overlap of significant genes across the three cohorts might reflects substantial inter-dataset heterogeneity: the TCGA cohort is primarily derived from a U.S. population, whereas the two CGGA datasets originate from Chinese patients. Differences in genetic background and cohort characteristics likely contribute to variations in prognostic gene signatures. From the subset of 41 genes that were consistently significant in all three cohorts, we selected MS4A6A for further investigation based on its strong and specific expression in macrophages, a cell type that dominates the immunosuppressive microenvironment of GBM, and its consistently robust prognostic association across all datasets. This combination of biological plausibility and translational relevance made MSA4A6A a good candidate for mechanistic follow-up. We acknowledge that several of the other intersecting genes are also compelling, and they may warrant future investigation. In fact, in parallel work we have also studied RNF135, another macrophage-associated immune gene, which further supports our interest in myeloid-centered regulatory mechanisms in GBM.
- For figure 2, author can clarify the origin of the data used in the plot. Also, for correlation plots, author can add the number of patients. For heatmaps (figure 2D-E), it would be nice to include the expression of MS4A6A as a part of annotation. Also, the color code for two heatmap should be consistent.
Response:
We thank the reviewer for these helpful suggestions. All analyses in Figure 2, including the ESTIMATE scores (A–C), xCell heatmap (D), MCP-counter results (E), and ssGSEA correlations (F), were generated using the TCGA-GBM bulk RNA-seq dataset (n=159). We have now added the exact sample size to the figure legends. In addition, the heatmaps have been updated to include unified color schemes, and the annotations have been standardized accordingly. The revised figure and legends have been incorporated into the manuscript.
- For Figure 5C, the compounds mentioned in cnotext in general have a lower drug sensitivity and difference between high and low groups is minimal. Author should justify why the small difference could make biological difference in clinical application. Also the plot needs to be rearranged or highlight the selected terms to make it easy to read.
Response:
We sincerely appreciate the reviewer’s valuable comment. We agree that the differences shown in Figure 5C appears modest in absolute terms. In the OncoPredict framework, the y-axis represents the predicted IC50, with a lower IC50 indicating greater drug sensitivity. This clarification has now been added to the figure legend to prevent any potential misunderstanding. It is also important to emphasize that this drug-response analysis was conducted primarily as a hypothesis-generating exercise—not as direct evidence of clinical efficacy. While the absolute differences in predicted IC50 are not large, they consistently suggest that MS4A6A-high tumors may exhibit heightened sensitivity to inhibitors targeting mTORC1/2, ERK1/2, and CDK4/6 pathways. These observations are intended to highlight potential therapeutic vulnerabilities for further investigation, and their clinical relevance will naturally require subsequent functional validation and preclinical studies. We have clarified this intent in the revised manuscript. We have also improved the readability of Figure 5C by adding boxed annotations to highlight compounds with higher predicted sensitivity in the MS4A6A-high group.
We added this in the limitation part of the manuscript: “In addition, the predicted drug sensitivity differences derived from OncoPredict are hypothesis-generating and will require further validation through pharmacological assays in experimental models.”

Round 2
Reviewer 2 Report
Comments and Suggestions for Authors
The revised manuscript has addressed the issued and is in good shape for publication